# Coupling Policy Gradient with Population-based Search (PGPS)

## Abstract

Gradient-based policy search algorithms (such as PPO, SAC, or TD3) in deep reinforcement learning (DRL) have shown successful results on a range of challenging control tasks. However, they often suffer from deceptive gradient problems in flat or gentle regions of the objective function. As an alternative to policy gradient methods, population-based evolutionary approaches have been applied to DRL. While population-based search algorithms show more robust learning in a broader range of tasks, they are usually inefficient in the use of samples. Recently, reported are a few attempts (such as CEMRL) to combine gradient with a population in searching optimal policy. This kind of hybrid algorithm takes advantage of both camps. In this paper, we propose yet another hybrid algorithm, which more tightly couples policy gradient with the population-based search. More specifically, we use Cross Entropy Method (CEM) for population-based search and Twin Delayed Deep Deterministic Policy Gradient (TD3) for policy gradient. In the proposed algorithm called Coupling Policy Gradient with Population-based Search (PGPS), a single TD3 agent, which learns by a gradient from all experiences generated by population, leads a population by providing its critic function Q as a surrogate to select better performing next generation population from candidates. On the other hand, if the TD3 agent falls behind the CEM population, then the TD3 agent is updated toward the elite member of CEM population using loss function augmented with the distance between the TD3 and the CEM elite. Experiments in five challenging control tasks in a MuJoCo environment show that PGPS is robust to deceptive gradient and also outperforms the state-of-the-art algorithms.

## 1 Introduction

In Reinforcement Learning (RL), an agent interacts with the environment, and its goal is to find the policy that maximizes the objective function, which is generally defined as a cumulative discounted reward. Recently, many researchers have worked on combining deep neural networks and a gradient-based RL algorithm, generally known as Deep Reinforcement Learning (DRL). This approach has achieved great success not only in the discrete action domain, such as in Go (Silver et al., 2017) and Atari games (Mnih et al., 2015; 2016), but also in the continuous action domain, such as in Robot control (Fujimoto et al., 2018; Lillicrap et al., 2015; Schulman et al., 2015). However, it is difficult to use the gradient-based method for the objective function ($J$),

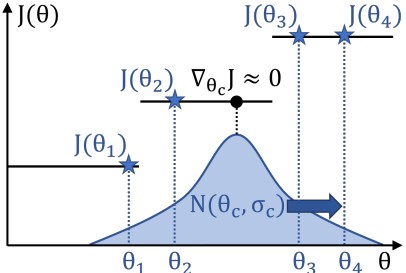

Figure 1: Flat gradient and population-based search on piece-wise constant function

which includes "many wide flat regions" since the gradient ($\nabla_\theta J$) is near zero at a flat point. Figure 1 is an extreme case consisting of only flat regions, which is called a piece-wise constant function. This problem remains an unsolved issue in gradient-based DRL with continuous control domains (Colas et al., 2018). The Swimmer in a MuJoCo environment (Todorov et al., 2012) has already been reported to be hard to use the gradient-based method (Jung et al., 2020; Liu et al., 2019). Our experiment shows that the objective function of Swimmer includes wide flat regions (Appendix A).

The population-based Evolutionary Approach (EA), which is an alternative to the gradient-based method, has also shown successful results in various control tasks (Conti et al., 2018; Liu et al., 2019;

Salimans et al., 2017; Such et al., 2017). As a population-based search, the EA generates a population of agents to explore policy, and the population is regenerated with improvement in each generation. The EA is also known as a direct policy search (Schmidhuber & Zhao, 1998) because it directly searches by perturbing the parameter of policy. In Figure 1, the Cross-Entropy Method (CEM) as a kind of population-based search is simply described, where the current population sampled from the target distribution is evaluated. Then the distribution is updated to the direction for generating a more promising population. Not depending on the gradient, these approaches are robust to flat or deceptive gradients (Staines & Barber, 2013; Liu et al., 2019). However, the EA is sample inefficient because it requires a Monte-Carlo evaluation, and the previous results and data generally cannot be reused.

The off-policy Policy Gradient (PG) algorithm uses the data from arbitrary policies to train its actor and critic functions. It generates exciting potential by combining the EA and PG, where the data which is discarded in a standard EA is directly used to train the PG's functions. Khadka & Tumer (2018) and Pourchot & Sigaud (2018) introduced a framework combining the EA and off-policy PG. However, the framework of (Khadka & Tumer, 2018) is less efficient to train the policy for general tasks than the PG algorithm alone, and the framework of (Pourchot & Sigaud, 2018) is unsuitable to train the policy for a task providing a deceptive gradient.

In this paper, we propose another hybrid algorithm, called Policy Gradient with Population-based Search (PGPS) in which the CEM and Twin Delayed Deep Deterministic Policy Gradient (TD3) (Fujimoto et al., 2018) are combined. It is as robust to a deceptive gradient as the CEM and more efficient to train the policy for general tasks than TD3. To be robust to a deceptive gradient, the proposed algorithm is constructed in a way similar to the one in (Khadka & Tumer, 2018), where the TD3 is trained using data from the CEM and periodically participates in the CEM population as an individual (PG guides EA). However, in this basic framework, the TD3 sometimes falls into the inferior solution and inefficiently searches. To get the TD3 out of the inferior solution, we let the EA guide the TD3 by guided policy learning (Jung et al., 2020) (EA guides PG). Furthermore, the TD3 critic contributes to generating a more promising population by filtering the set of actors sampled from CEM (Q-critic filtering). Lastly, to control the trade-off between the frequency of search and stable estimation, we used evaluation step scheduling in the process of population evaluation (Increasing evaluation steps). It carries out frequent searches when searching far from the optimal, whereas it carries out stable estimation when searching close to the optimal. These approaches bring out more synergies between the CEM and the TD3 while maintaining both the population-based search and the gradient-based search. Consequently, the proposed algorithm is not only robust to a deceptive gradient, but also produces outstanding performances with a low additional computational cost.

## 2 RELATED WORKS

Recently, beyond the view of an alternative approach, few attempts have been proposed in the form of A supporting B. An attempt is to use EA to fill a replay buffer with diverse samples. In Colas et al. (2018), a Goal Exploration Process (GEP), a kind of EA, is firstly applied to search the policy and to fill a replay buffer with the diverse samples, and then the off-policy PG algorithm is sequentially used for fine tuning the parameters of the policy. Another attempt is to combine a population-based approach and PG for efficiently searching a good policy or the good hyper-parameters of an algorithm in parallel multi-learners setting. These applications generally consist of periodically evaluating the population, followed by distributing good knowledge to the other learners. To find the best architecture and hyper-parameters, Jaderberg et al. (2017) proposed a Population-Based Training (PBT) method in which the current best knowledge is periodically transferred to PG learners. Gangwani & Peng (2017) developed the distilled crossover using imitation learning and mutation based on the PG. Proposed operators transfer the information on current good policies into the next population without destructive change to the neural network. Jung et al. (2020) introduced a soft-manner guided policy learning to fuse the knowledge of the best policy with other identical multiple learners while maintaining a more extensive search area for the exploration.

The idea of combining the population-based EA and off-policy PG was recently introduced by Khadka & Tumer (2018). Their approach was called Evolutionary-Guided Reinforcement Learning (ERL) in which the Genetic Algorithm (GA) and the Deep Deterministic Policy Gradient (DDPG) (Lillicrap et al., 2015) are combined. In ERL frameworks, the GA transfers the experience from evaluation into the DDPG through a replay buffer, and the DDPG transfers the knowledge learned

from the policy gradient into the GA by periodically injecting a PG actor into the GA population. Khadka et al. (2019) expanded the PG algorithm of ERL from a single DDPG learner to multiple TD3 learners with a resource manager. Bodnar et al. (2019) revised the GA's crossover and mutation to the distilled crossover and proximal mutation inspired by (Gangwani & Peng, 2017) and (Lehman et al., 2018) to prevent the destruction of neural networks. Pourchot & Sigaud (2018) introduced another framework, which combines the CEM and the TD3. In this framework, the TD3 algorithm has only a critic function trained using the experience from the CEM. In order to propagate the knowledge learned by policy gradient to the CEM, half of the population is updated to the direction indicated by the TD3 critic for a fixed number of steps, followed by the evaluation. The policy gradient for half of the population not only enhances the gradient-based learning, but also deteriorate the CEM's robustness over a deceptive gradient.

In this paper, we introduce another hybrid algorithm, in which the CEM and the TD3 are combined as in CEMRL (Pourchot & Sigaud, 2018). However, the TD3 has both actor and critic, which are trained by a gradient from experiences generated by CEM. And then, the TD3 actor periodically participates in CEM population as in ERL (Khadka & Tumer, 2018). This structure is an effective way to maintain a direct policy search of CEM. To enhance the performance, we introduced new interactions processes between the CEM and TD3 instead of carrying out a policy gradient for numerous individual actors.

## 3 BACKGROUNDS

**Twin Delayed Deep Deterministic Policy Gradient (TD3)**   RL framework has an agent interacting with an environment generally defined by a Markov Decision Process (MDP). At each timestep $t$, an agent receives the state $s_t$, and takes an action $a_t$ according to the policy $\pi$, and then receives a reward $r_t$ and the next state $s_{t+1}$ at next time step $t+1$. The goal of RL is to find the policy that maximizes the discounted cumulative return $R_t = \sum_{k=t}^{\infty} \gamma^{k-t} r_k$ where $\gamma$ is a discounted factor. Off-policy RL can use the data from arbitrary policies to train its actor and critic functions repeatedly, which is a key point for improving recent gradient-based RL. Silver et al. (2014) introduced the off-policy Deterministic Policy Gradient (DPG), which has an advantage for high-dimensional action spaces. The DDPG (Lillicrap et al., 2015) was extended from the DPG to apply it to a deep neural network. TD3 (Fujimoto et al., 2018) is an advanced version of the DDPG, which suffers from the overestimation bias of the critic. To correct this bias, two critics are introduced, and then the critic with the lowest state-action value is taken during the critic update as in the Double Deep Q-Network (DDQN) (Van Hasselt et al., 2016). Figure 2(a) represents the architecture of the TD3.

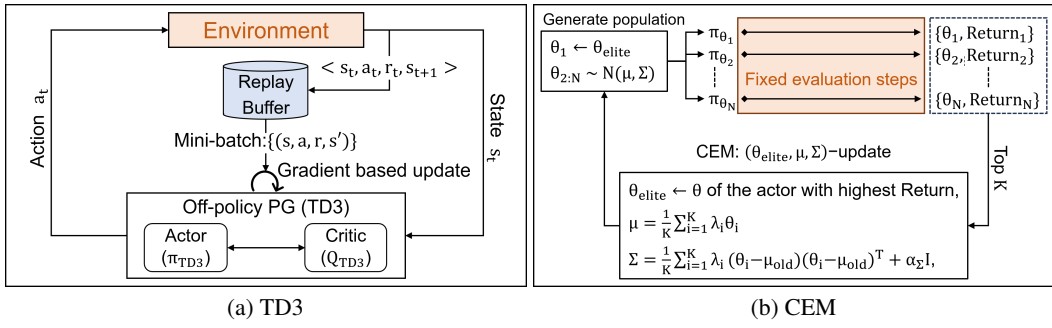

(a) TD3                (b) CEM

Figure 2: Architecture of TD3 and CEM. $\theta_i$ is sampled from $N(\mu, \Sigma)$, $\lambda_i$ is a weight depending on the rank of the $Return_i$, and $K$ is a fixed number about high performing actors.

**Cross Entropy Method (CEM)**   The Evolutionary Approach (EA) is a heuristic search method inspired by nature, where the current population is evaluated, and the next population is regenerated using the current evaluation result in order to produce a higher Return, which is also known as Fitness and defined as a cumulative sum of immediate reward for a fixed number of steps. The Estimation of Distribution Algorithm (EDA) is a class of the EA: It updates the target distribution to generate a better population. Depending on the update method for the distribution, EDAs are classified as a CEM (De Boer et al., 2005), a Covariance Matrix Adaptation Evolutionary Strategy (Hansen, 2016), an Evolutionary Strategy (Salimans et al., 2017), and a Natural Evolutionary Strategy (Wierstra et al., 2014). We used the CEM as one part of our proposed algorithm. As shown in Figure 2(b), the CEM procedures are as follows: The population is sampled from the multivariate Gaussian $N(\mu, \Sigma)$ and

evaluated, and for the top $K$, which is smaller than the population size ($N$), high performing actors are used to update a new mean ($\mu$) and covariance ($\Sigma$) of the target distribution. The weight can be given to each actor according to the rank of the Return (Hansen, 2016). The elite actor can be passed to the next population, which is known as elitism. The more detailed procedure is reported in (De Boer et al., 2005). In this paper, we used a diagonal covariance to reduce the parameters.

**Population-guided Parallel Policy Search (P3S)** The Guided Policy Learning (GPL) is commonly used when the elite policy leads some sub-policies to a better solution in multiple policies setting. Teh et al. (2017) introduced GPL for joint learning of numerous tasks in which a common policy encourages local policies to act better. Jung et al. (2020) proposed a soft-manner GPL, called the Population-guided Parallel Policy Search, for multiple identical learners with the same objective, where a population is evaluated periodically. Then sub-policies are trained to maximize their critic value and to concurrently minimize the distance from the elite policy for the next period. For this purpose, Augmented Loss (2) is used to train the sub-policies instead of Original Loss (1).

$$Original\ Loss: \quad L^O(\pi) = E_{s\sim SS}[-Q_\pi(s, \pi(s))] \tag{1}$$

$$Augmented\ Loss: L^A(\pi, \pi_{elite}, \beta) = E_{s\sim SS}[-Q_\pi(s, \pi(s)) + \beta||\pi(s) - \pi_{elite}(s)||_2^2] \tag{2}$$

where $\pi$ is a trained policy, $Q_\pi$ is a critic function depending on $\pi$, $\pi_{elite}$ is the elite policy, $SS$ is the set of states, and $||\pi(s) - \pi_{elite}(s)||_2^2$ is the Euclidean distance measure between the trained policy and the elite policy. $\beta$ is a distance weight and is controlled adaptively. In this paper, we used a revised GPL inspired by P3S so that the CEM elite actor guides the TD3 to better space.

# 4 COUPLING POLICY GRADIENT WITH POPULATION BASED SEARCH ALGORITHM

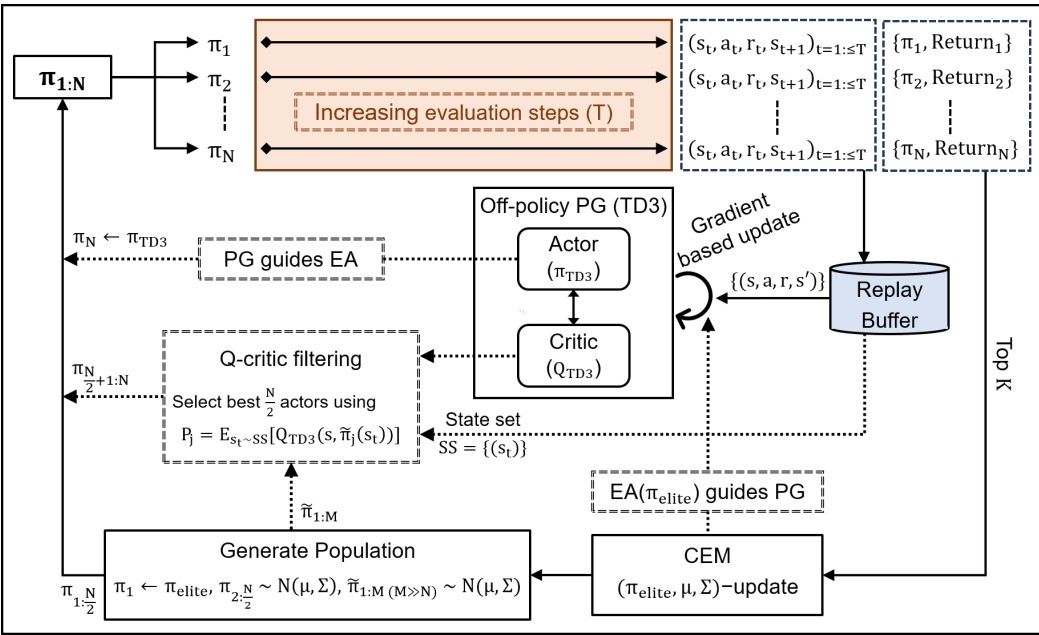

Figure 3: Architecture of PGPS. $\pi$ is used instead of $\pi_\theta$ for the simplification.

As shown in Figure 3, the Policy Gradient with Population-based Search (PGPS) is a coupling framework between the CEM and the TD3. In this framework, two algorithms encourage each other to be efficient. The general flow is as follows. The generated actor population is evaluated by interacting with the environment, where each actor can fail to reach the max step ($T$). The experience (the set of state transitions) is saved in a replay buffer, and Returns go to the CEM. The CEM updates the parameters of the population using top $K$ ($K$ is set to half of the population) high performing actors, and then the TD3 trains its critic and actor using the mini-batches sampled from the replay buffer. The knowledge of TD3 is periodically transferred to the CEM by copying the TD3 actor

into the last actor of the CEM population. The knowledge of the CEM is transferred to the TD3 by the GPL when the TD3 is presumed to falls into the inferior solution. Lastly, the next population is regenerated, where the TD3 critic is used to selects promising actors among the set sampled from $N(\mu, \Sigma)$ so that the population will be better. A pseudocode of the PGPS is described in Appendix B.

**Gradient-based Update**   In a standard CEM, the experience of state transitions is discarded immediately because they require only Returns to update the target distribution. However, the TD3 enables the discarded experience to be reused to train its actor and critic functions. Therefore, the experience is saved in the replay buffer and then is repeatedly used in a gradient-based update.

**PG Guides EA**   In order to transfer the knowledge learned in TD3 to the CEM, the TD3 actor is periodically copied to the last actor of the population. If the last actor is included in the top-performing $K$ actors, a multivariate Gaussian move to the direction indicated by TD3. On the other hand, if the TD3 actor is excluded from the top-performing $K$ actors, the CEM ignores the knowledge from the TD3 and focuses on the direct policy search. High copying frequency benefits knowledge propagation from the TD3 to the CEM, but it can disturb the population-based search.

**EA Guides PG**   When the TD3 falls into the inferior solution, mainly due to a deceptive gradient, it is difficult to get out by relying solely on the gradient-based method despite the experience from good behavior policy (Colas et al., 2018). Therefore, we use the Guided Policy Learning (GPL), where the CEM elite actor leads the TD3 to escape the inferior solution. We judge the TD3 to have fallen into the inferior solution if its actor shows a lower Return than *(mean − one standard deviation)* of Returns of the current population. The TD3 actor that has fallen into the inferior solution is updated for the direction to minimize the Augmented Loss $L^A$, equation (2). Moreover, the TD3 critic is trained through the target actor indirectly guided by the elite actor, and it will fix the critic to be appropriate. The distance weight ($\beta$) in the in $L^A$ is adapted several times during the GPL-based update by equation (3). It is a simplified version of P3S (Jung et al., 2020) and similar to Adaptive TRPO, which was introduced in (Schulman et al., 2017).

$$\beta = \begin{cases} \beta \ \times \ 2 & \text{if } D(\pi_{TD3}, \pi_{elite}) > D_{target} \ \times \ 1.5 \\ \beta \ / \ 2 & \text{if } D(\pi_{TD3}, \pi_{elite}) < D_{target} \ / \ 1.5 \end{cases} \tag{3}$$

where the distance measure $D(\pi_{TD3}, \pi_{elite})$ is defined as $E_{s \sim SS}[||\pi_{TD3}(s) - \pi_{elite}(s)||_2^2]$, $SS$ is the set of states in the replay buffer, and $D_{target}$ as a hyper-parameter determines how close the TD3 actor and the elite actor should be. During the GPL-based update, the TD3 actor stays around the CEM elite actor while maximizing its critic value.

**Q-critic Filtering**   It is important to generate a good population, since it not only leads $N(\mu, \Sigma)$ to be better but also encourages the TD3 to be trained well by filling the replay buffer with good experience. However, one cannot know which actor is better before the evaluation. To estimate the potential in advance, we use the Q-critic as a surrogate model (Jin, 2011). It can sort out promising actors from the set of actors sampled from $N(\mu, \Sigma)$ before the evaluation.

**Proposition1** If $E_{a \sim \pi_i(\cdot|s)}[Q_{TD3}(s, a)] \geq E_{a \sim \pi_{TD3}(\cdot|s)}[Q_{TD3}(s, a)]$ *for all* $s$,
$$\mathrm{E}_{a \sim \pi_i(\cdot|s)}[Q_{\pi_i}(s, a)] \geq E_{a \sim \pi_{TD3}(\cdot|s)}[Q_{TD3}(s, a)]$$

Specific proof is described in Appendix C. We assume that a higher $E_{a \sim \pi_i(\cdot|s)}[Q_{TD3}(s, a)]$ indicates a higher $E_{a \sim \pi_i(\cdot|s)}[Q_{\pi_i}(s, a)]$. In this assumption, $E_{s \sim SS}[Q_{TD3}(s, \pi_i(s))]$ is used as the relative potential of the actor $i$ ($\pi_i$) where SS is the set of states in the replay buffer. The overall procedure is as follows: $M$ ($\gg$N) actors are sampled from $N(\mu, \Sigma)$, and then Q-critic fills half of the population with the actors with higher relative potential by filtering M actors. The remaining half consists of the elite actor and the actors sampled from $N(\mu, \Sigma)$ for the exploration.

**Increasing Interaction Steps**   In order to efficiently control the trade-off between the frequency of searches and stable performance estimation of actors, we used evaluation steps scheduling, where the evaluation steps between actor and environment increase with cumulative evaluation step by equation (4).

$$T = min(T_{init} + 100 \times mod(cumulative \ evaluation \ steps, T_{inter}), T_{max}) \tag{4}$$

where $T$ is the current evaluation step, which means that each actor can maximally interact with the environment as much as $T$. $T_{init}$ is the initial evaluation step, $T_{inter}$ is the interval for increasing evaluation step, and $T_{max}$ is the maximum evaluation step depending on the task.

The evaluation step should be sufficiently long for stable performance estimation of the population, but it reduces the number of the CEM generation and delays the update of the TD3. Short evaluation steps make it possible to carry out more population-based searches and frequent TD3 updates but causes unstable performance estimation. As the arbitration approach, we used the increasing evaluation steps. This approach guarantees more searches and frequent updates at the beginning stage of learning and fine estimation at the later stage of learning.

## 5 Experiments

### 5.1 Comparison to Baselines

**Environment**    The proposed algorithm (PGPS) was empirically evaluated on five games in Mujoco managed by OpenAI Gym, which is widely used as continuous control benchmarks.

**Baseline Algorithms**    We compared the performance of PGPS with various EA, PG, and EAPG algorithms, such as CEM, PPO(Schulman et al., 2017), DDPG(Lillicrap et al., 2015), TD3(Fujimoto et al., 2018), SAC(Haarnoja et al., 2018), CERL(Khadka et al., 2019), PDERL(Bodnar et al., 2019), and CEMRL(Pourchot & Sigaud, 2018). Almost algorithms were implemented using the code provided by its authors. However, the DDPG is implemented by the code provided by the authors of the TD3. OpenAI Spinningup (Achiam, 2018) was used to implement PPO. PGPS and CEM were implemented using PyTorch. Our code is available at http://github.com/NamKim88/PGPS.

**Hyper-parameter Setting**    For stable learning, we performed tuning on the architecture of the neural network and the hyper-parameters of the learning rate, population size, the period of copying TD3 actors to the population, increasing evaluation steps, and Q-critic filtering. The detailed setting is described in Appendix D. Adam (Kingma & Ba, 2014) was used to train the neural networks in the TD3. The hyper-parameters of baseline algorithms were set as the same as the reference code.

**Performance Evaluation Metrics**    Each algorithm learns five times at different seeds for a task. Each learning runs for a million timesteps. That is, the total numbers of interacting all agents with the environment are a million. The evaluation tests are performed every 10,000 steps. Each evaluation test performed without any exploration behavior and reports the average reward over ten episodes, where the evaluation step of a episode is 1,000. For the evaluation test of the PGPS, the current mean ($\mu$) of $N(\mu, \Sigma)$ was used as the parameters of the evaluation policy. If the learning is performed at seed 1, the evaluation test will proceed at random seed 1 + 100. This approach is the same as the original code of the TD3 (Fujimoto et al., 2018) and applied to all baseline algorithms. The curves in Figure 4 reported over the average performance of policies trained at five random seeds from 1 to 5.

**Results**    In Figure 4 the performance of all baseline algorithms is similar to that reported in the original author's papers and reference papers. Some differences come from the training seed, evaluation metrics, the variance of algorithms, and the version of MuJoCo and Pytorch.

The results show that all PG algorithms suffer from a deceptive gradient in Swimmer-v2. In contrast, the CEM, which is a direct policy search method, yields the best result over a deceptive gradient in Swimmer-v2. The CERL and PDERL as a combining algorithm between GA and PG show better performance than PG algorithms. However, their performances are lower than the CEM because the knowledge propagated from the PG to the GA disturbs the direct policy search of the GA. Despite the CEMRL combining algorithm between the CEM and TD3, it shows similar performance to the TD3. This result comes from that the gradient-based update of half of the population ruins the population-based search of the CEM.

In the remaining four tasks which are advantageous to the gradient-based method, advanced PG algorithms show much better performance than the CEM. The performances of CERL and PDERL are located between an advanced PG algorithm and the CEM in most cases. Especially, CERL yields lower performance than the TD3, which is one part of CERL. This is due to the failure of GA and

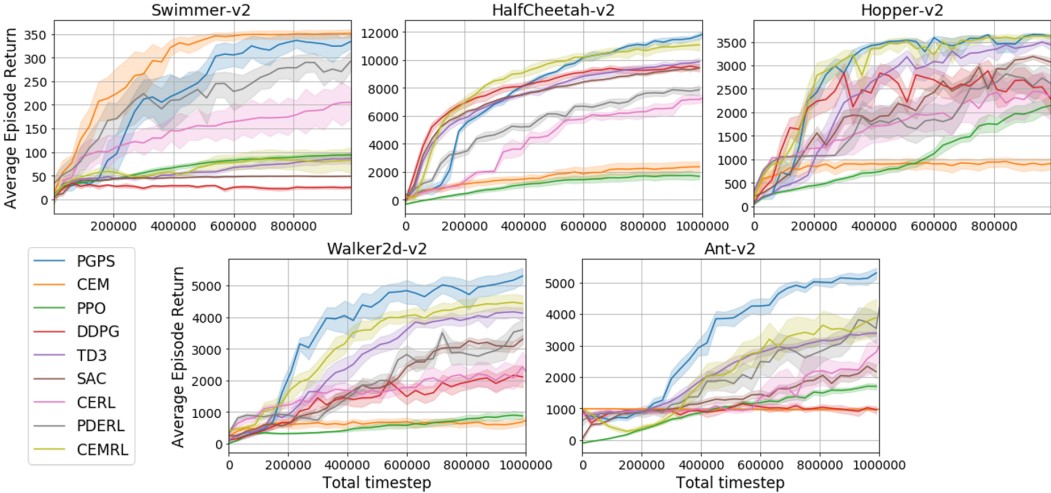

Figure 4: Learning curves obtained on 5 continuous control tasks.

TD3 to combine effectively. CEMRL outperforms all baseline algorithms in four tasks since the gradient-based update for multiple actors amplifies the advantage of the gradient method.

PGPS carried out gradient-based update on the TD3, and then the TD3 actor is periodically copied to the last actor of the CEM population. It keeps the computational cost lower and also minimizes the disturbance from the gradient method on a direct policy search. As a result, the PGPS can achieve results comparable to the CEM in Swimmer-v2 which is advantageous to a direct policy search. Furthermore, the PGPS shows an outstanding performance in the remaining four tasks, which are advantageous to the gradient-based method. That performance is due to the additional interaction processes for coupling two algorithms efficiently, such as mutual guidance (PG-EA), Q-critic filtering, and increasing evaluation steps.

## 5.2 ABLATIONS STUDIES

In this subsection, we performed ablations studies to investigate the effect on the final performance and computational time when a specific process is cumulatively added to the base model. The added sequence is as follows: PG guides EA (P to E), EA guides PG (E to P), Q-filtering, and increasing evaluation steps. HalfCheetah-v2, Walker2d-v2, and Ant-v2 are used for the ablation studies. Each task was learned five times at random seeds of 1 to 5. Each learning runs for a million timesteps. Table 1 reports the average over 15 (three games×five learning) runs. Evaluation test is performed using the $\mu$ of the CEM except for the Base model.

**Base model**  The experience from CEM is saved to a replay buffer. However, the TD3 and CEM are trained independently without any guidance. Two evaluation tests are performed using the TD3 and the $\mu$ of the CEM at the end of learning. A better one is selected for the final performance.

Table 1: Ablation studies on 3 continuous control tasks.

| | Model | Average normalized score | Normalized computational time | | | | |
|---|---|---|---|---|---|---|---|
| | | | Sum | TD3 update | Actors evaluation | Population generation | CEM update |
| (1) | Base model | 1.00 | 1 | 0.934 | 0.065 | 0.001 | $\approx 0$ |
| (2) | (1) + P to E | 3.73 | 1 | 0.934 | 0.065 | 0.001 | $\approx 0$ |
| (3) | (2) + E to P | 4.26 | 1.107 | 1.041 | 0.065 | 0.001 | $\approx 0$ |
| (4) | (3) + Q-filtering | 4.44 | 1.134 | 1.041 | 0.065 | 0.028 | $\approx 0$ |
| (5) | (4) + Increasing | 5.26 | 1.134 | 1.041 | 0.065 | 0.028 | $\approx 0$ |

P to E shows the most noteworthy performance improvement within the proposed algorithm. A good experience is essential to train TD3's actor and critic functions well. As the TD3 actor directly guides the population to be better, the population fills the replay buffer with good experience, and then it

makes sure the TD3 will be trained well again. For the perspective of the CEM, the TD3 actor is a good exploration, which cannot be developed from a random sampling, and improves the population to be better. For the perspective of the TD3, the experience from the population is a richer set of exploration behavior, which is better than PG-based action space exploration (Plappert et al., 2017). The other processes also additionally contribute to performance improvement of about $4 \sim 18\%$ The remarkable point is that these processes incur a low additional computational cost. The Q-filtering requires a relatively highest computational cost, but it is only about $11\%$ of that of the TD3 alone.

**The effect of EA guidance**   In contrast to the existing hybrid algorithm (Bodnar et al., 2019; Khadka et al., 2019) in which the PG only guides the EA, the proposed algorithm lets the EA also guides the TD3. It is beneficial to pull the TD3 agent out of the inferior solution, especially if it is captured due to a deceptive gradient. Figure 5 represents the effect of EA guidance (EA guides PG) in swimmer-v2. As shown in Figure 5, the TD3 actor without EA guidance stays on the inferior solution most of the time, whereas the TD3 actor with EA guidance quickly gets out of the inferior solution and goes on searching for a better space. The proposed algorithm will also perform robust learning

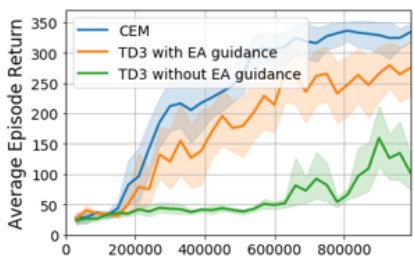

Figure 5: Learning curves of TD3 with(out) EA guidance on Swimmer-v2

even if both deceptive and general gradient occurs in the environment since It lets CEM lead TD3 when a deceptive gradient occurs and lets TD3 lead CEM when the general gradient occurs.

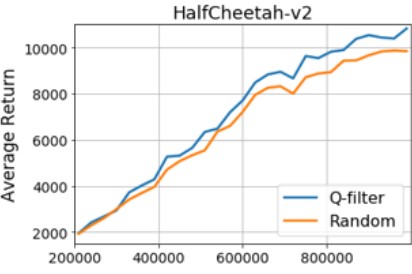

Figure 6: Average Return of filtered actors and sampled actors during running

**The effect of Q-critic filtering**   The proposed algorithm selects half of the population using Q-critic (Q-filter), and the remaining half is sampled from a multivariate Gaussian (Random). Figure 6 shows the average Return of Q-filter and Random in HalfCheetah-v2. Q-filter shows a $7.7\%$ higher average performance than Random. (Because the evaluation steps were controlled, the average Return in Figure 6 is lower than the test episode return in Figure 4.) This result is additional evidence that Q-critic filtering is meaningful for sorting out promising actors. The remaining concern is the effect of Q-critic filtering on the exploration. If Q-critic selects similar actors, it might deteriorate the exploration behavior for both the EA and the PG. To further improve exploration, the distance-based criteria introduced in (Bodnar et al., 2019) can also be used with Q-critic filtering.

## 6   CONCLUSIONS

In this paper, we proposed another hybrid algorithm coupling the Cross-Entropy Method (CEM) and the Twin Delayed Deep Deterministic Policy Gradient (TD3), which is called Policy Gradient with Population-based Search (PGPS). Proposed algorithm is not only robust to a deceptive gradient, which is difficult to be learned by the TD3 alone, but it also achieves higher sample efficiency, which is deficient in the EA alone. To be robust to a deceptive gradient with low additional computational cost, we revised the existing hybrid algorithm framework to make it an improved structure. Also, to enhance the performance, we introduced new interaction processes such as mutual guidance (PG↔EA), Q-critic filtering, and increasing evaluation step. First, mutual guidance is the most crucial process, where the TD3 guides the CEM to make it better, and the CEM also guides the TD3 that has fallen into the inferior solution to make it better. Second, Q-critic helps the population to consist of more promising actors by filtering the set of actors sampled from a multivariate Gaussian. Lastly, the increasing evaluation step controls the trade-off between the frequency of searches and stable estimation. It takes the frequent searches and updates when searching on a coarse policy space far from the optimal at the beginning stage of learning and fine estimation close to the optimal at the later stage of learning. Our experiments on a MuJoCo confirmed that the proposed algorithm outperforms state-of-the-art PG and hybrid algorithms, while its computational cost is kept at about 13.5% higher than the TD3 algorithm.

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

# A    APPENDIX A. THE SHAPE OF SWIMMER-V2

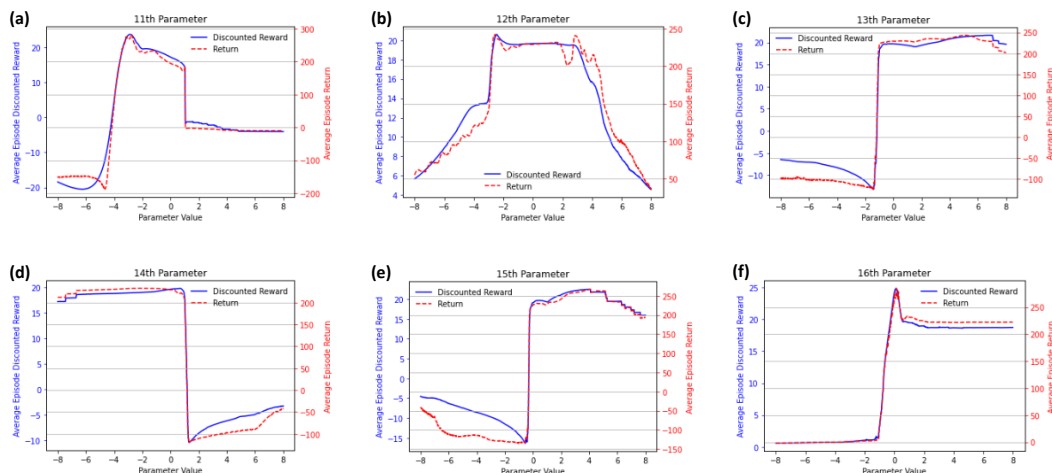

Figure 7: The graphs represent the change of average episodic discounted reward  average episodic return of a linear policy actor by the value change of a single parameter in the Swimmer, where the other parameters were fixed and we evaluated thirty times at different seeds for each policy.

Average Episode Discounted Reward = $E_{\pi(\cdot|s)}[\sum_{t=0}^{\infty}\gamma^t r_t^\pi] \approx \frac{1}{30}\sum_{n=1}^{30}\sum_{t=0}^{999}\gamma^t r_t^\pi$

Average Episode Return = $E_{\pi(\cdot|s)}[\sum_{t=0}^{\infty} r_t^\pi] \approx \frac{1}{30}\sum_{n=1}^{30}\sum_{t=0}^{999} r_t^\pi$

where $\gamma = 0.99$, and the architecture of the linear policy : [state dim, action dim] $\rightarrow$ tanh

To find the reason why the previous state-of-the-art policy gradient methods, such as TRPO, PPO, SAC, and TD3, were hard to solve the Swimmer in the Mujoco Environment, we performed an interesting two-steps experiment. In the first step, we sat an actor, following a simple linear policy, and executed the CEM algorithm to find an optimal policy parameters θ in the Swimmer. When the Return value of the actor reached over 200, we saved the policy parameters $\theta^2 00$. In the next step, while changing one parameter value, we evaluated the changed policy thirty times with different seeds and recorded the all Discounted Rewards ($J(S_0)$) and Returns. We represents the some cases of the experiment in Figure 1. As you can find in the Figure 1, we can discover interesting facts: 1) on the graphs (a), (d), and (f) in Figure 1, we can find wide regions, where the gradients $\nabla_\theta J(S_0)$ are near zero; 2) except for (b) in Figure 1, at particular parameter value, the gradient of $J(S_0)$ and Return is steep enough to be seen as a piece-wise; and 3) on the graphs (c), (d) and (e), the graphs are shaped like valleys neat at those steep points. We suspected the above facts as the cause of the deceptive gradient problem of the Swimmer, and raised the question in the Introduction. Finally, by considering those issues, we propose the combined algorithm with TD3 and CEM.

# B  APPENDIX B. PSEUDOCODE OF PGPS ALGORITHM

---

**Algorithm 1** Coupling Policy Gradient with Population based Search Algorithm

---

**Set hyper-parameters:**  TD3: $lr_{actor}, lr_{critic}, \tau, \gamma$, and $\kappa$ ; CEM: pop-size $N$, top $K$, $\Sigma_{init}, \Sigma_{end}$, and $\tau_{\Sigma}$ ; Mutual guidance: $F_{TD3 \to CEM}, \beta_{init}$, and $D_{target}$ ; Q-critic filtering: $T_{start-Q}$ and $SR$ ; Increasing interaction steps: $T_{max}, T_{init}$, and $T_{interval}$

1: Initialize the mean $\mu$ of the multivariate Gaussian of the CEM
2: Initialize the TD3 actor $\pi_{TD3}$ and TD3 critic $Q_{TD3}$
3: initialize replay buffer $R$
4: total_steps = 0
5: **for** generation=1:$\infty$ **do**
6:    **if** total_steps $\geq$ T$_{start-Q}$ **then**
7:       $pop \leftarrow$ Q-critic Filtering($N, \pi_{elite}, \mu, \Sigma, Q_{TD3}, R$)
8:    **else**
9:       $pop[1] \leftarrow \pi_{elite}, pop[2:N]$ are sampled from $N(\mu, \Sigma)$
10:    **end if**
11:    **if** generation mod $F_{TD3 \to CEM} = 0$ **then**
12:       $pop[N] \leftarrow \pi_{TD3}$
13:    **end if**

14:    $T = min(T_{init} + 100 \times mod(total\_steps, T_{interval}), T_{max})$

15:    interaction_steps = 0
16:    **for** i=1:pop_size $N$ **do**
17:       Set the current actor $\pi$ as $pop[i]$.
18:       $Return_i, (s_t, a_t, r_t, s_{t+1})_{t=1:t_{end}(t_{end} \leq T)} \leftarrow$ Evaluate($\pi, T$)
19:       Fill replay buffer $R$ with $(s_t, a_t, r_t, s_{t+1})_{t=1:t_{end}}$
20:       interaction_steps = interaction_steps + $t_{end}$
21:    **end for**
22:    total_steps = total_steps + interaction_steps

23:    Update ($\pi_{elite}, \mu, \Sigma$) with the top-$K$ Return actors

24:    num_update = interaction_steps / 5
25:    **if** generation mod $F_{TD3 \to CEM} = 0$ and $Return_N <$ MEAN($Returns$)-STD($Returns$) **then**
26:       **for** i=1:5 **do**
27:          Sampled states 2 ($SS_2$) are drawn from $R$
28:          Update $\beta = \begin{cases} \beta \times 2 & \text{if } E_{s \sim SS_2}[||\pi_{TD3}(s) - \pi_{elite}(s)||_2^2] > D_{target} \times 1.5 \\ \beta \ / \ 2 & \text{if } E_{s \sim SS_2}[||\pi_{TD3}(s) - \pi_{elite}(s)||_2^2] < D_{target} \ / \ 1.5 \end{cases}$
29:          Train $Q_{TD3}$ for num_update mini-batches from $R$ using a standard TD3 algorithm
30:          Train $\pi_{TD3}$ for num_update mini-batches from $R$ to minimize $L^A(\pi_{TD3}, \pi_{elite}, \beta)$
31:       **end for**
32:    **else**
33:       **for** i=1:5 **do**
34:          Train $Q_{TD3}$ for num_update mini-batches from $R$ using a standard TD3 algorithm
35:          Train $\pi_{TD3}$ for num_update mini-batches from $R$ to minimize $L^O(\pi_{TD3})$
36:       **end for**
37:    **end if**
38: **end for**

---

In contrast to a standard TD3(Fujimoto et al., 2018) performing one step interaction with environment and then one update repeatedly, our TD3 carries out as many updates as the sum of the evaluation steps of the current generation after the evaluation. In proposed algorithm, the total updates steps is divided into 5 iterations. At each iteration, a critic firstly trained for a fixed number of steps followed by an actor is trained for same steps. If total update steps are 10,000, at each iteration, critics is firstly trained for 2,000 minibatches, and then actor is trained for 2,000 minibatches for the direction

to maximize critic. For stabilizing the volatility of critic, it is widely used in the implementation (Achiam, 2018; Hill et al., 2018; Pourchot & Sigaud, 2018). $L^O(\pi_{TD3})$ is original TD3 Loss in equation (1). $L^A(\pi_{TD3}, \pi_{elite}, \beta)$ is augmented loss for guided policy learning in equation (2).

---

**Algorithm 2** Function Q-critic Filtering

---

1: **procedure** Q-critic Filtering($N, \pi_{elite}, \mu, \Sigma, Q_{TD3}, R$)
2:     $pop[1] \leftarrow \pi_{elite}, pop[2 : N/2]$ are sampled from $N(\mu, \Sigma)$
3:     $\pi_{j=1:M(=SR*N)}$ are sampled from $N(\mu, \Sigma)$
4:     Sampled states 1 ($SS_1$) are drawn from replay buffer $R$
5:     **for** j=1:M **do**
6:         $P_j = E_{s \sim SS_1}[Q_{TD3}(SS_1, \pi_j)]$
7:     **end for**
8:     $pop[N/2 + 1, N] \leftarrow$ Select policies ($\pi$s) with higher $P_j$ among $\pi_{j=1:M}$
9:     **Return** $pop$
10: **end procedure**

---

**Algorithm 3** Function Evaluate

---

1: **procedure** Evaluate($\pi_{elite}, T$)
2:     $returns, t$, buffer $(BF) = 0, 0, [\,]$
3:     Reset environment and get initial state $s_0$
4:     **while** env is not done **and** t $\leq$ T **do**
5:         Select action $a_t = \pi(s_t)$
6:         Execute action $a_t$ and receive reward $r_t$ and next state $s_{t+1}$
7:         Fill $BF$ with stat transition $(s_t, a_t, r_t, s_{t+1})$
8:         $returns = returns + r_t$ and $t = t + 1$
9:     **end while**
10:     **Return** $returns, BF$
11: **end procedure**

---

In a standard TD3 algorithm, Gaussian noise or Ornstein-Uhlenbeck(Uhlenbeck & Ornstein, 1930) noise are added to the action $a_t$ for exploration. It is usually known as action space noise. Pourchot & Sigaud (2018) empirically showed that action space noise does not contribute the performance improvement. We also cannot find any evidence for the advantage about action space noise. Therefore, proposed algorithm does not use action space noise.

## C APPENDIX C. PROOF OF PROPOSITION 1

In this section, we prove Proposition 1.

**Proposition1** If $E_{a \sim \pi_i(\cdot|s)}[Q_{TD3}(s,a)] \geq E_{a \sim \pi_{TD3}(\cdot|s)}[Q_{TD3}(s,a)]$ for all $s$,
$$E_{a \sim \pi_i(\cdot|s)}[Q_{\pi_i}(s,a)] \geq E_{a \sim \pi_{TD3}(\cdot|s)}[Q_{TD3}(s,a)]$$

Proof. For arbitrary $s_t$

$V_{\pi_{TD3}}(s_t)$

$= E_{a_t \sim \pi_{TD3}(\cdot|s_t)}[Q_{\pi_{TD3}}(s_t, a_t)]$

$\leq E_{a_t \sim \pi_i(\cdot|s_t)}[Q_{\pi_{TD3}}(s_t, a_t)]$

$= E_{a_t \sim \pi_i(\cdot|s_t)}[r_t^{\pi_i} + \gamma E_{a_{t+1} \sim \pi_{TD3}(\cdot|s_{t+1})}[Q_{\pi_{TD3}}(s_{t+1}, a_{t+1})]]$

$\leq E_{a_t \sim \pi_{TD3}(\cdot|s_t)}[r_t^{\pi_i} + \gamma E_{a_{t+1} \sim \pi_i(\cdot|s_{t+1})}[Q_{\pi_{TD3}}(s_{t+1}, a_{t+1})]]$

$= E_{a_t \sim \pi_i(\cdot|s_t)}[r_t^{\pi_i} + \gamma r_{t+1}^{\pi_i} + \gamma^2 E_{a_{t+2} \sim \pi_{TD3}(\cdot|s_{t+2})}[Q_{\pi_{TD3}}(s_{t+2}, a_{t+2})]]$

$\leq E_{a_t \sim \pi_i(\cdot|s_t)}[r_t^{\pi_i} + \gamma r_{t+1}^{\pi_i} + \gamma^2 E_{a_{t+2} \sim \pi_{TD3}(\cdot|s_{t+2})}[Q_{\pi_{TD3}}(s_{t+2}, a_{t+2})]]$

$\cdots$

$\leq E_{a_t \sim \pi_i(\cdot|s_t)}[r_t^{\pi_i} + \gamma r_{t+1}^{\pi_i} + \gamma^2 r_{t+2}^{\pi_i} + \cdots + \gamma^\infty r_\infty^{\pi_i} + \cdots]$

$\cong E_{a_t \sim \pi_i(\cdot|s_t)}[\sum_{k=t}^\infty \gamma^{k-t} r_k^{\pi_i}]$

$\cong E_{a \sim \pi_i(\cdot|s)}[Q_{\pi_i}(s,a)]$

$= V_{\pi_i}(s_t)$

We assumed that higher $E_{a \sim \pi_i(\cdot|s)}[Q_{TD3}(s,a)]$ means higher $E_{a \sim \pi_i(\cdot|s)}[Q_{\pi_i}(s,a)]$. Therefore, the policy $\pi$ with higher $E_{a \sim \pi(\cdot|s)}[Q_{TD3}(s,a)]$ is better. We used sampled state ($SS$) from repay buffer to estimate the performance of policy. That is, $E_{s \sim SS}[Q_{TD3}(s, \pi(s))]$ is a estimator for $E_{a \sim \pi(\cdot|s)}[Q_{TD3}(s,a)]$ for all s. Sum up, the policy with higher $E_{s \sim SS}[Q_{TD3}(s, \pi_i(s))]$ is better policy.

# D    APPENDIX D. DETAILED HYPERPARAMETERS SETTING

Table 2 includes the architecture of the neural networks. Table 3 represents the hyperparameters that kept constant across all tasks. Table 4 describes the hyperparameters that vary with the task.

Table 2: The architecture of the neural networks.

| Actor | Critic |
|---|---|
| [state dim, 400] | [state dim + action dim, 400] |
| elu | elu |
| [400, 300] | [400, 300] |
| elu | elu |
| [300, action dim] | [300, 1] |
| tanh | - |

Table 3: Hyperparameters constant across all tasks.

| Hyperparameter | Value |
|---|---|
| Target weight ($\tau$) | 0.005 |
| TD3 Actor learning rate ($lr_{actor}$) | 2e-3 |
| TD3 Critic learning rate ($lr_{critic}$) | 1e-3 |
| Discount factor ($\gamma$) | 0.99 |
| Replay buffer size | 1e-6 |
| Batch size ($\kappa$) | 256 |
| CEM initial covariance ($\Sigma_{init}$) | 7.5e-3 |
| CEM limit covariance ($\Sigma_{limit}$) | 1e-5 |
| Initial distance weight ($\beta_{init}$) | 1 |
| Target distance ($D_{target}$) | 0.05 |
| Sampled states 2 ($SS_2$) size | 512 |
| Q-filtering start step ($T_{start-Q}$) | 150000 |
| Multiple sample ratio ($SR$) | 50 |
| Sampled states 1 ($SS_1$) size | 64 |
| Max interaction ($T_{max}$) | 1000 |
| Initial interaction steps ($T_{init}$) | 400 |
| Interval for increasing step ($T_{inter}$) | 100000 |

Table 4: Hyperparameters varying across tasks.

| Task | Swimmer | HalfCheetah | Hopper | Walker2d | Ant |
|---|---|---|---|---|---|
| Population size (N) | 10 | 10 | 6 | 6 | 6 |
| top K | 5 | 5 | 3 | 3 | 3 |
| Frequency of TD3 to CEM ($F_{TD3 \to CEM}$) | 3 | 1 | 1 | 1 | 1 |
| Decaying covariance constant ($\tau_\Sigma$) | 0.01 | 0.03 | 0.03 | 0.03 | 0.03 |

