# OpenReview forum: "PGPS : Coupling Policy Gradient with Population-based Search"
_ICLR.cc/2021/Conference — Reject_

### Official Review · AnonReviewer3 · 2020-10-28
**An implementation that merges two known methods**

**Rating:** 5
**Confidence:** 4

**Review:**

This a nicely written paper.  The authors propose an algorithm that combines population-based search with policy gradient approach to obtain optimal policies in reinforcement learning. The proposed combination leads to a yet another algorithm aiming at exploitation (TD3) and exploration (CEM). The idea of using such combinations is not new (a point also acknowledged by the authors). The results are promising, though these results are obtained with some additional computational cost. I believe this manuscript should be considered as an implementation work, where the novelty of the idea is not clear.

---

### Official Review · AnonReviewer2 · 2020-10-29
**Combination of existing algorithmic components, evaluation is not strong enough**

**Rating:** 5
**Confidence:** 3

**Review:**

Summary:

Recently, several researchers have been trying to combine the goodnesses of direct policy search approaches (mostly based on evolutionary computation approaches)  and those of policy gradient approaches in control tasks. This paper proposes a novel combination of an evolutionary direct policy search and an actor-critic approach. The authors combines a cross-entropy method, which directly samples parameters of actor network (policy) from a Gaussian distribution that is trained during the search process, and the twin delayed deep deterministic policy gradient (TD3), which is an off-policy actor-critic approach.

In existing combination of an evolutionary direct policy search and an actor critic approach, the evolutionary search part contributes to fill the replay buffer with diverse experiences, whereas the actor critic approach leads the population of the evolutionary search by replacing subset of the population with the policy of the actor critic agent, which is often expected to be superior to the population of the evolutionary search. Differently from these existing researches, the proposed approach also make an elite population to guide the actor critic agent when it is stacked at an area where the policy gradient almost vanishes. This idea comes from the recently proposed P3S-TD3, another population based (multi-policy) TD3 approaches.

Comparison has been conduced on 5 MuJoCo environments. In Swimmer-v2 environment, the proposed approach reached the performance between the pure cross entropy method (best) and other combinations of evolutionary approaches and policy gradient approaches. In the other four environments, the proposed approaches reached the competitive or superior average performance.

Criticism:

Compared to the results presented in P3S-TD3 paper (ICLR 2020), which used a slightly different environment and therefore the direct comparisons may not be valid, differences observed in this paper between the proposed approach and the pure TD3 is rather similar to the differences between P3S-TD3 and TD3. Since one of the main contributions of this paper, if I understand correctly, is the introduction of the component borrowed from P3S-TD3, P3S-TD3 should be included in the baseline approaches in the comparison. Is there any advantage of the proposed approach over P3S-TD3 on HalfCheetah, Hopper, Walker2d and Ant?

Swimmer environment is the only tested environment where CEM works significantly better than TD3. On this environment the proposed approach was shown to be as effective as CEM. Does it generalize to other environment where CEM works significantly better than TD3? The current evaluation lacks the evaluation of the generality of this approach.

In the ablation study (Table 1) the effect of Equation (4) was evaluated, and shows a non-negligible impact. Equation (4), which controls the number of interaction steps performed in each MC evaluation in CEM, can be incorporated to other approaches such as CEMRL, and if the same impact appears in the existing approaches, it is ambiguous whether the other components really contribute to improve the performance over existing approaches.

---

### Official Review · AnonReviewer1 · 2020-11-01
**The paper title references the achievement of the closely-followed CEM-RL.**

**Rating:** 3
**Confidence:** 4

**Review:**

**Summary:** The authors propose PGPS as a minor variation on [CEM-RL](https://arxiv.org/abs/1810.01222).

**Quality:** The quality of the paper is overall very low: misleading and unsustained claims, poor and incorrect literature review, cherry-picking results.

**Clarity:** While the exposition overall follows a standard template, the writing is in dire need of intensive copy efforts prior to being considered for publication.

**Originality:** The authors present a minor augmentation of CEM-RL while claiming some of its results as their own.

**Pros:**
- The experimental part seems to have received a lot of work
- The setup and hyperparameter description is very complete, coupled with open sourced code, potentially allowing full experiment reproduction

**Cons:**
- The first sentence of the Conclusions claims the pairing of CEM and TD3 integrating policy gradient and population-based search as a novelty and contribution, while it was introduced in the (cited!) CEM-RL.
- There is no clear, explicit definition of the paper's contributions, which given the circumstances is unacceptable.
- Some results (particularly regarding the Q-critic filtering) are sketchy and not statistically meaningful.
- The Related Work study in Section 2 is riddled with incorrectness in the evolutionary part. For example: stating EA stands for Evolutionary Approach rather than Algorithms; referring to CMA-ES by citing a 2016 tutorial instead of the [2001 paper](https://scholar.google.ch/scholar?q=completely+derandomized); equating EA to direct policy search; stating that previous-generation data cannot be reused; and many more. This overall shows a less than perfunctory understanding of the field. The parallels with the CEM-RL paper are once again obvious and misquoted.
- The paper writing requires a thorough overhaul.

**Comments:**
The contributions are not made clear, and a quick read of the original CEM-RL paper shows an exact correspondence but for 1. the double-feedback loop and 2. the Q-critic filtering. These two contributions however are very minimal, and most importantly they are not tested in isolation: end of Section 5, 1. "The effect of EA guidance" does not compare with CEM-RL (or other hybrid methods), and 2. "The effect of Q-critic filtering" drops the variance information in favor of what looks like single runs with statistically indistinguishable results.

**Final remarks:**
This work is a barely-augmented, error-riddled rewriting of a 2019 paper, and its state is nowhere near publishable. I strongly advise the authors to decide if their contributions are sufficient for publication, and if so to write a paper dedicated to proving their point to the scientific community.
Finally, the un-anonymized GitHub link on Section 5.1 breaks the code of conduct; the fact that the author (NamKim88) is less known does not excuse the breach of anonymity.

---

> ### Comment · ~Jiaming_Cheng1 · 2024-08-23
> **stating that previous-generation data cannot be reused**
>
> EAs typically suffer with high sample complexity and often struggle to solve high dimensional problems that require optimization of a large number of parameters. The primary reason behind this is EA’s **inability to leverage powerful gradient descent methods which are at the core of the more sample-efficient DRL approaches.** Moreover, the replay buffer utilized by off-policy is a vital component in enabling sample-efficient learning. However, a significant problem of the population from ERL is that it does not directly exploit the individual experiences collected by the actors in the population.

---

> ### Comment · ~Jiaming_Cheng1 · 2024-08-23
> **stating EA stands for Evolutionary Approach rather than Algorithms**
>
> A large number of literature claims that EA refers to evolutionary algorithms

---

### Official Review · AnonReviewer5 · 2020-11-04
**Experiments not convincing**

**Rating:** 5
**Confidence:** 4

**Review:**

The paper proposes a new method combining evolutionary methods and RL. In particular, the authors combine CEM and TD3 in PGPS. PGPS maintains a population of policies, which interact with the environment to collect data filling the replay buffer. The data in replay buffer is then used to train TD3. PGPS enables information flow in both directions: when the TD3 policy performs poorly, the elite policy from the population is used to guide TD3 by an imitation learning loss; The TD3 critic helps select top policies in the population and the TD3 actor is also included in the population. The experiments on simple Mujoco domains demonstrate the utility of PGPS and the ablation study analyzes the utility of each part of PGPS.

My major concern is that the experiments are not convincing enough.
(1) I think TRES (Liu et al., 2019), Salimans et al. (2017), and PBT (Jaderberg et al., 2017) are important baselines that the authors need to compare with.
(2) The reported results are based on only 5 random seeds which I don't think are enough.
(3) The experiments have only 5 domains, I think at least Humanoid should be included. Moreover, as PBT does demonstrate advantages in video games (e.g., Atari, DMLab), I'd like to see if PGPS can scale up to video games as well.

---

### Comment · ~Namyong_Kim1 · 2023-03-01
**Thank you for your helpful comments and insightful suggestions**


25 Nov 2020 (modified: 25 Nov 2020)ICLR 2021 Conference Paper1654 Official CommentReaders: Program Chairs, Paper1654 Area Chairs, Paper1654 Reviewers, Paper1654 Authors
Comment:
To. All reviewers

We really want to thank you for your helpful comments and insightful suggestions. We will try to improve our proposed algorithm and experiments following your review comments.

To reviewer 1

• We did not recognize the video game environment. Thank you.

• Following your suggestion, we will perform additional experiment with more random seeds and TRES and PBT as additional baselines.

To reviewer 2

• Thank you for pointing out our shortcoming specifically.

• We will revise the conclusion and try to mention our contribution more clearly.

• For double looped feedback and Q-critic filtering, we will try to refine our work.

To reviewer 3

• Actually, we have the result of running P3S-TD3 code in the same environment (halfcheetah-v2, hopper-v2, walker2d-v2, ant-v2, and swimmer-v2) by utilizing the code of P3S-TD3 authors. We will summarize that result and add it to our paper.

• We are not aware of generality of our approach. We will try to consider more about it. Thank you.

• To more clearly show the effect of components of our algorithm, we will try to perform more ablation study.

To reviewer 4

• We will try to describe the contribution of our algorithm and research more clearly by referring to our observation in Introduction section. Thank you.

---

### Decision · Program_Chairs · 2021-01-07
**Final Decision**

**Decision:**

Reject

**Comment:**

This paper proposes a hybrid algorithm that combines RL and population-based search. The work is interesting and well-written. But, the contribution of the work is very limited, in comparison with the state-of-the-art.